# Simultaneous Quantitation and Discovery (SQUAD) Analysis: Combining the Best of Targeted and Untargeted Mass Spectrometry-Based Metabolomics

**DOI:** 10.3390/metabo13050648

**Published:** 2023-05-10

**Authors:** Bashar Amer, Rahul R. Deshpande, Susan S. Bird

**Affiliations:** Thermo Fisher Scientific, San Jose, 95134 CA, USA; rahul.deshpande2@thermofisher.com (R.R.D.); susan.bird@thermofisher.com (S.S.B.)

**Keywords:** metabolomics, simultaneous quantitation and discovery (SQUAD), targeted metabolomics, untargeted metabolomics

## Abstract

Untargeted and targeted approaches are the traditional metabolomics workflows acquired for a wider understanding of the metabolome under focus. Both approaches have their strengths and weaknesses. The untargeted, for example, is maximizing the detection and accurate identification of thousands of metabolites, while the targeted is maximizing the linear dynamic range and quantification sensitivity. These workflows, however, are acquired separately, so researchers compromise either a low-accuracy overview of total molecular changes (i.e., untargeted analysis) or a detailed yet blinkered snapshot of a selected group of metabolites (i.e., targeted analysis) by selecting one of the workflows over the other. In this review, we present a novel single injection simultaneous quantitation and discovery (SQUAD) metabolomics that combines targeted and untargeted workflows. It is used to identify and accurately quantify a targeted set of metabolites. It also allows data retro-mining to look for global metabolic changes that were not part of the original focus. This offers a way to strike the balance between targeted and untargeted approaches in one single experiment and address the two approaches’ limitations. This simultaneous acquisition of hypothesis-led and discovery-led datasets allows scientists to gain more knowledge about biological systems in a single experiment.

## 1. Introduction

Many biological fields require the accurate prediction of phenotype from genotype. This has led to sophisticated work for the analysis of genomes (i.e., genomics) as well as the study of their expressed transcriptomes (i.e., transcriptomics). Proteomics and metabolomics are also crucial to gain information that enables a wider understanding of both function and phenotype [1]. Many years ago, scientists came to realize that the flow of biochemical information is a set of interactions between the different omics fields, shown in Figure 1, rather than directional from the genome to the metabolome [2]. Therefore, a multi-omics integration is necessary to elucidate biological systems’ chemical structure, function, development, adaptation, and evolution for a deeper understanding of the principles of life [1].

Metabolomics, as a family member of “omics”, is the measurement of small molecule substrates, intermediates, and/or end products of cellular metabolism (i.e., metabolites). It provides an immediate and dynamic response to genetic and/or environmental perturbations in biological systems including biofluids, cells, tissues, and/or organisms [3]. Mass spectrometry (MS)-based analyses coupled to a separation technique (e.g., liquid chromatography (LC), gas chromatography (GC), or capillary electrophoresis (CE)) are dominantly used in metabolomics. Matrix-assisted laser desorption/ionization (MALDI)-MS metabolomics is also used in some cases since it can conduct high-throughput screening without the need for separation. Nuclear magnetic resonance (NMR) spectroscopy, in addition, is a powerful analytical technique for high-throughput metabolic fingerprinting that provides reliable metabolite structure (e.g., via 2D NMR) identification [4].

The diverse physiochemical properties of the metabolome limit our ability to analyze all metabolites from a biological system with a single or even a limited set of analytical techniques [3]. In addition, the choice of method for a metabolomics study depends on the research question, the complexity of the sample, and the available resources. However, multiple methods are used, when feasible, for comprehensive metabolome characterization.

When implemented to study biological processes and systems, metabolomics can be a valuable technique for biomarker discovery, disease diagnosis, and biochemical pathway elucidation, especially when used to measure the metabolic differences between unperturbed and perturbed groups, such as healthy control and patients with a particular disease [5].

Figure 2 presents the trends in “omics” techniques utilization during the period 2000–2022 based on the number of annual publications obtained from a search in PubMed (https://pubmed.ncbi.nlm.nih.gov/ (accessed on 1 March 2023)) using title and title/abstract search factors. The figure shows steady growth in the number of genomics publications during the last 22 years, which might be due to the advancement in DNA sequencing, resulting in reduced cost and increased throughput [6]. Transcriptomics also appears to show a relatively moderate increase in the number of publications during the last two decades (Figure 2). According to this search, proteomics utilization has been following a similar trend as genomics. Interestingly, the search also suggests a significantly growing trend in the application of metabolomics-based research starting in 2010 making it to lead the trend among other “omics” family members for the last year. This could be explained by the fact that there is a growing need for more phenotypic information. Consequently, scientists are using these “omics” techniques to facilitate their research. The improvements in proteomics and metabolomics analytical capabilities may also have contributed to the potential growth in their utilization.

Traditionally, metabolomics studies are performed using either an untargeted approach or a targeted approach, which are discussed in this review. Both untargeted and targeted metabolomics have their strengths and weaknesses. The greatest challenge for the untargeted, for example, is maximizing the detection and accurate identification of thousands of metabolites, while for the targeted, the greatest challenge is maximizing the linear dynamic range and quantification sensitivity [5].

## 2. Untargeted Metabolomics

Untargeted metabolomics aims to determine all measurable metabolites in a biological sample. The untargeted analyses are often used for biomarker discovery and are hypothesis-generating approaches that focus on acquiring data for as many compounds as possible, annotating metabolites, and reviewing both known and unknown metabolic changes in the studied system. Data from the untargeted approach are normally used for relative quantification across sample groups that could be further validated with targeted analysis [7].

### 2.1. Workflow

Effective metabolite extraction from a given sample is essential for the success of untargeted metabolomics. Such a method must liberate a broad range of metabolites from the sample that are normally diverse in their physiochemical properties [8]. Full scan MS^1^ followed by either a data-dependent acquisition (DDA) or data-independent acquisition (DIA) is the common workflow for data acquisition in untargeted metabolomics. These approaches aim to generate accurate mass measurements for individual features to allow the confident annotation of compounds (obtained from the high-resolution accurate MS^1^ and informative MS^2^ spectra) and statistical calculations (e.g., differential analysis, obtained from the MS^1^ full scan). 

In contrast to DIA, the mass spectrometer in DDA mode selects only certain analytes (i.e., the highest in abundance) and then fragments them, ideally one at a time. DDA is the preferred method for library generation and database searches, however, since the triggering of precursor ions for MS^2^ analysis is intensity dependent, some metabolic features with low MS abundance may never be selected for fragmentation. This can lead to missing the annotation of metabolites of interest [9]. In the DIA approach, MS^2^ spectra are generated for all precursor ions in small m/z ranges, thus detecting and measuring low-abundance metabolites. Yet, the link between precursors and their fragments is dissociated due to the complexity of the resulting MS^2^ spectra, which increases the challenge of MS^2^ spectra deconvolution for metabolite annotation [10]. 

Quality assurance (QA) and quality control (QC) processes are required for ensuring analytical performance and reliability for untargeted metabolomics studies. QA comprises procedures done prior to data acquisition to test that the analytical system is suitable to obtain the required data quality. QC includes procedures done during and immediately after the sample analyses to test that the analytical data are reliable and reproducible. QA/QC strategies, therefore, improving the integrity and robustness of untargeted metabolomics datasets and saving time and valuable samples by obviating the need for repeated analysis. QA activities include securing thoughtful experimental design, proper staff training, and construction of standard operating procedures (SOPs). On the other side, QC strategies include the analysis of pooled QC samples for intra-study assessment, system suitability standards and blanks, internal standards and/or standard reference materials for correction, normalization, and assessment.

### 2.2. Limitations

The diverse chemical properties and concentrations of metabolites in any biological sample complicate their extraction, separation, and detection in an untargeted metabolomics fashion. This makes researchers compromise on a combination of analytical parameters (e.g., stationary phase for separation and ionization mode for MS-based detection), which may improve the detection of some substances but reduce the detection of others. Another solution would be the combination of different extraction and separation methodologies, which increases the time of analysis and requires solutions to merge information from different analytical techniques (e.g., reversed-phase and HILIC-based MS). Another challenge in untargeted metabolomics, apart from maximizing the detection, is securing accurate identification of thousands of metabolites. Moreover, untargeted analysis can suffer from signal bias and mass drift resulting from the analysis of complex sample matrices, which reduces overall sensitivity. Thus, it is crucial to use high-resolution accurate-mass (HRAM) detection, such as that provided by Orbitrap mass analyzers, when feasible, for identifying and determining the elemental and isotopic contents of a sample with high levels of precision. In addition, there is still a need for advanced pathway analysis tools to interpret metabolomics data to solve some of the most challenging biological paradoxes [2]. Finally, the data generated by untargeted metabolomics are usually very complex and require sophisticated statistical and bioinformatics tools for data analysis and interpretation.

## 3. Targeted Metabolomics

Targeted metabolomics aims to measure defined groups of chemically characterized and biochemically annotated metabolites [8]. This analysis requires a more hypothesis-driven type of experiment, where there is a recognized rationale for selecting the metabolite group under study [11]. The preferred analytical technologies for an ideal targeted metabolomic analysis should provide a combination of high sensitivity, specificity, linear dynamic range, and throughput. Targeted metabolomics is often used for biomarker validation, where the aim is to confirm the differential expression of specific metabolites that have been identified using untargeted metabolomics. MS-based techniques combined with a separation technique such as LC and GC are widely used in targeted metabolomics routines. A targeted LC-MS approach was used to determine concentration levels of bile acids, short-chain fatty acids (SCFAs), and tryptophan/indole metabolites in mouse feces and/or cecal contents in a study that aimed to rationally design bacterial consortia to treat chronic immune-mediated colitis and restore the intestinal homeostasis [12]. Meanwhile, a GC-MS-based targeted metabolomics was used for the identification of highly sensitive biomarkers that can aid the early detection of pancreatic cancer [13]. Further, some targeted metabolomics kits can quantify up to 500 compounds [14,15].

Metabolite quantitation in this manner has different levels of accuracy. The most accurate is the absolute quantitation of targets using authentic standards (analyzed in a dilution series) and internal standards (i.e., isotopically labeled compounds). Labeled internal standards (IS) are typically spiked into the sample’s matrix and the calibration dilutions of the authentic standards at the same concentrations expected in the biological sample. These labeled IS provide ionization normalization through gradient coelution with the compound of interest. A less accurate quantitation is done using calibration curves of authentic standards only, and finally, a semi-quantitation protocol uses a one-point calibration strategy, which is quick and easy to apply but provides less accurate data. Because many metabolites have no commercially available IS, either isotopically labeled or unlabeled, researchers have investigated using one IS per metabolite or lipid class, compromising accuracies in comparison to truly quantitative methods. Nevertheless, such quantifications are still more reliable, and (more importantly), comparable between different laboratories and studies than untargeted metabolomics [16]. The capability to perform accurate quantitative experiments is the gold standard of targeted metabolomics [17]. The authentic chemical standards of the studied metabolites are also used for accurate identification (level 1 identification using RT, MS fragmentation patterns, peak shape, and high-resolution accurate mass of molecular ion) in targeted metabolomics. 

### 3.1. Workflow

Ensuring a proper design of MS-based targeted metabolomics is crucial since it can significantly affect the total number of measured metabolites and the reproducibility of experimentation. This includes the optimization of sample preparation, which includes determining the concentration of each IS to be spiked into the sample matrix (for absolute quantitation) and selecting the right extraction and clean-up procedures. Effective metabolite extraction should focus on the extraction of the metabolite class of interest since only a subset of metabolites will be analyzed. Therefore, the extraction method should be tailored to the physicochemical properties and relative abundances of the specifically targeted metabolites while not including components such as macromolecules (e.g., proteins) and nonrelevant metabolites. Extraction parameters include monophasic or biphasic liquid-liquid extraction, volume and ratio of the organic and aqueous solvents used, and the pH and temperature at which the extraction is carried out [8].

Based on the physicochemical properties of targets, HILIC and reversed-phase LC and GC methodologies are optimized for the separation of analytes before MS detection to reduce ion suppression effects. Within the mass spectrometer, different ways of treating the ions formed during the ionization can be used in targeted analysis experiments. Analyte fragmentation is required as part of some targeted metabolomic analyses to increase targeted specificity. Such fragmentation can take place during ionization (in-source fragmentation) or by selecting ions and specifically fragmenting them by causing collisions between the gaseous ions and an inert gas (i.e., tandem MS) [11]. Multiple reaction monitoring (MRM) and selected reaction monitoring (SRM) experiments use the signal of selected MS^2^ fragment ions for quantitation and are widely used in targeted metabolomics. They are typically performed on triple-quadrupole mass analyzers. While SRM is monitoring only a single fixed mass window, MRM scans rapidly over multiple (very narrow) mass windows and thus acquires traces of multiple fragment ion masses in parallel. It is the application of SRM to multiple product ions from one or more precursor ions. Other experiments such as parallel reaction monitoring (PRM) are also implemented for targeted metabolomics. While MRM/SRM monitors each precursor ion/product ion transition at a time, PRM analyzes all product ions derived from a precursor ion in concert with high resolution and mass accuracy. Both MRM/SRM and PRM allow relative and absolute quantification of metabolites. Finally, although HRMS is commonly used in untargeted metabolomics; it could also be used in targeted metabolomics [5], this is attributed to the improvements to the orbitrap technologies regarding scan speed, ultrahigh resolution, and sensitivity [18,19]. Using HRAM mass spectrometry, targeted analysis can be performed using PRM with the benefit of greater selectivity compared to full scan or selected ion monitoring methods, confirmation of data with MS/MS, retroactive data analysis, and less method development.

It is crucial to employ QC measures throughout any targeted metabolomics study to enhance the reliability of data and increase confidence in the results [20]. Reference standard-based QC samples are used to assess system (e.g., LC-MS) accuracy and stability. These QC samples are usually distributed among the batch sequence at regular intervals so their accuracy and coefficient of variance can be used to determine measurement robustness and identify (and possibly correct for) drift effects within the batch [20,21].

### 3.2. Limitations

Regardless of the advantages of targeted metabolomics, it has limited coverage of the metabolome. This can lead to missing information from biologically relevant metabolites that are outside of the predefined subset. In addition, it is difficult to obtain all the required pure chemical standards and IS for the metabolites of interest, therefore the coverage of detected metabolites in targeted metabolomics is generally limited [5].

## 4. Simultaneous Quantitation and Discovery (SQUAD) Analysis

As mentioned earlier in this review, both untargeted and targeted approaches have their strengths and weaknesses when applied to metabolomics research. Researchers, therefore, compromise either a low-accuracy overview of total molecular changes (i.e., untargeted analysis) or a detailed yet blinkered snapshot of a selected group of metabolites (i.e., targeted analysis). However, a relatively new workflow, SQUAD analysis, which combines both untargeted and targeted workflows, has recently emerged as a middle ground, merging the advantage of the two traditional approaches, and addressing their limitations (see Figure 3). This technique involves the quantification of a pre-defined set of metabolites, as in targeted metabolomics, but also allows for the identification of new metabolites that were not included in the targeted list.

The primary focus of this approach is the confident annotation (via authentic chemical standards) and the accurate quantitation (via calibration curves using authentic standards and isotopically labeled standards) of a targeted set of metabolites. In addition, the secondary focus is to find new molecular connections in the system by performing untargeted analysis, via retro-mining recorded data, on a single injection. The quantitation pillar of this approach is flexible, ranging from absolute if both standards and IS are available to relative quantitation in the absence of IS and calibration curves (i.e., one-point calibration). See Section 3 for more details on metabolite quantitation. 

In this review, we present a comprehensive single-injection approach that combines both targeted and untargeted analyses. We also recommend naming the new metabolomics workflow as Simultaneous Quantitation and Discovery (SQUAD) Analysis. In addition, we discuss how SQUAD metabolomics could offer the best of both worlds and how this technique can be implemented by laboratories to overcome barriers across multiple fields for their analytical needs.

### 4.1. Nomenclature

The combination of targeted and untargeted analyses has been reported in the literature in various forms that are not necessarily based on a single injection analysis, and this review aims to discuss these methods. In addition, the workflow has been given several names such as semi-targeted metabolomics [22], hybrid metabolomics [5], pseudo-targeted metabolomics [23], and combined targeted and untargeted metabolomics [24]. Interestingly, some of these names have been used for multiple workflows that are not necessarily the same. Semi-targeted metabolomics, on the one hand, was used to describe doing both targeted and untargeted metabolomics via single method injection [25]; on the other hand, it was also used to secure level 1 identification of the targeted compounds [22]. Moreover, the term “semi-targeted” can be misleading to the community since it does not imply fully combining the two traditional workflows in the analysis performed but that it is only done partially. To this end, we find it urgent to present an expressive term for this new workflow that can be used by the community without any confusion. Based on that, we propose the name Simultaneous Quantitation and Discovery (SQUAD) Analysis. The new workflow of combining targeted and untargeted metabolomics is based on a single injection method; therefore, it simultaneously enables quantitation and discovery.

### 4.2. Workflow Structures

In this section, we will discuss the most common workflow structures that are implemented to combine both targeted and untargeted metabolomics. In addition, we will present and discuss SQUAD analysis workflows. Interestingly, many researchers have been combining the two approaches in their work without identifying the workflow, while others have identified it as “combining the two approaches” or called it “semi-targeted” or “pseudo-targeted” metabolomics.

#### 4.2.1. Combined Targeted and Untargeted Metabolomics

In the last decade, researchers recognized the importance of merging the targeted and the untargeted approaches to have comprehensive and informative coverage benefiting from the accurate measurement of specific targets and the discovery aspect that allows for data retro-mining. In 2010, Pinel et al. discussed the potential of implementing profiling approaches (untargeted) as valuable tools for combating the illegal use of growth promoters in cattle breeding besides the traditional MS^2^ targeted approach. This is because the latter fails when faced with new xenobiotic growth-promoting agents or new methods of application, such as the administration of low-dose cocktails. In this context, screening strategies allowing the detection of the physiological response resulting from the administration of anabolic compounds are promising approaches to detect misuse [24]. The discussions at that time were about combining the outcome of the two approaches that are run on different platforms rather than one platform. 

Combining results from the two approaches was also used in the natural products field to determine the bioavailability of known and unknown biologically active phytochemicals [26,27]. Another methodology based on HRAM analysis was developed using targeted and untargeted screening strategies to discover potential biomarkers for the reliable detection of food product adulteration [28,29,30]. Many other studies from various research fields have also benefited from combining MS-based data from targeted and untargeted approaches [31,32,33,34,35,36,37,38,39,40,41] and even NMR-targeted data with MS-untargeted analysis [42]. Melnik et al. (2017), for example, developed a platform coupling targeted and untargeted metabolomics via splitting the flow from one UPLC into an orbitrap DDA-based discovery and a triple-quadrupole (TQ) MRM quantitation method to study the association between the metabolome and microbiome of human fecal samples [43]. 

Professor Fiehn demonstrated in 2016 an example of how to use GC-MS-based metabolomics for the integration of targeted assays for absolute quantification of specific metabolites with untargeted metabolomics to discover novel compounds in biological samples (i.e., blood, urine, cell culture, and homogenized tissue) [44]. External reference standards were used for the identification and absolute quantitation of targeted compounds, complemented by database annotations using large spectral libraries and validated standard operating procedures for the discovery and semi-quantification of non-targeted compounds in Fiehn’s study. In the same year, professor Fiehn and Cajka discussed in a review paper whether novel MS techniques, such as ultrahigh resolution detection, DDA MS^2^ fragmentation methods, and ion mobility, have advanced enough so that selectivity and sensitivity of untargeted analyses can enable the acquisition of hypothesis-driven validation studies on accurate mass profiling methods rather than classic TQ MRM [16]. They concluded that although advances in MS have had a big impact on overall metabolomics and lipidomics workflow, there is still a need for further improvement in metabolomics and lipidomics platforms for the merging of targeted and untargeted analyses. Remarkably, among the future directions they discussed is moving from HPLC to 2.1 mm UHPLC or 1 mm microflow LC chromatography to increase sample throughput and sensitivity. Other options include using advanced full spectra collection MS^1^ systems with an increased mass resolving power, enabling polarity switching or collection of both MS^1^ and MS^2^ (DDA or DIA) without significant loss of signal intensity. Additionally, there is an urgent need for databases with validated sets of MRM transitions for targeted metabolites in addition to larger libraries of MS^2^ spectra for compound identification in untargeted metabolomics and statistical tools to assess probabilities for the correct molecular annotation [16].

In his article, Dr. Rochat discussed why LC-HRMS (like the one an orbitrap can offer) will become a key instrument in clinical labs [45]. He highlighted that advanced HRMS instruments can perform sensitive and reliable quantifications of a large variety of analytes in MS^1^ full scan or, if needed, in a more targeted SIM or MS^2^ mode. Since these platforms record high-quality MS^1^ full scans, researchers can use those to get a global picture of a sample extract as virtually all ionized compounds are detected. Dr. Rochat expected that the use of one MS platform from targeted quantification to untargeted metabolomics will streamline workflows in the clinical environment after setting the basis in quantitative LC-HRMS analysis. He also added that only LC-MS analysis performed with HRMS allows efficient simultaneous Quan/Qual metabolomics analysis, which makes HRMS the instrument of choice for research projects [45].

Increased interest was shown by researchers to do both targeted and untargeted metabolomics in a single injection that can benefit situations with limited biological samples like the study of mouse optic nerves to study neurological visual diseases using GC-MS [46]. In 2018, Coene et al. presented a single LC-MS method that can be applied not only for holistic metabolic profiling in the plasma of individual suspected inborn errors of metabolism to detect and measure biomarkers, but also to simultaneously perform an untargeted assay [47]. They called this method “Next-generation metabolic screening”.

#### 4.2.2. Pseudo-Targeted Metabolomics

The pseudo-targeted metabolomics is another term that was used to describe the approach combining the advantages of untargeted and targeted analyses. Chen et al. proposed a pseudo-targeted approach to perform serum metabolomic analysis using a UHPLC/TQMS system operated in the MRM mode, for which the MRM ion pairs were acquired from the serum samples through untargeted tandem MS using UHPLC/Q-TOF MS/MS [23]. This approach was called pseudo-targeted since no standards or IS were used to determine the MRM transitions and to make calibration curves. A similar pseudo-targeted approach was later applied by Xu et al., who used a UPLC-HRMS (i.e., Q Exactive; Thermo Fisher Scientific) platform to perform untargeted analysis on urine to select targets and determine their MRM ion pairs before quantification on a QTRAP mass spectrometer coupled to LC. In this study, the MRM ion pairs were also determined by searching relevant databases [48]. Another similar analysis, which is called pseudo-targeted metabolomics, was developed for the identification and visualization of common pathogens by analyzing a QC sample (mix of different bacterial strains) on a hybrid Quadrupole-Orbitrap mass spectrometer to select analytes for quantitation via an MRM analysis on TQ MS [49]. Deng et al. also developed a pseudo-targeted approach to study metabolic changes in Asian plum (*Prunus salicina*) fruits in response to gummosis disease using a QTRAP for untargeted profiling followed by quantification of selected metabolites using MRM mode [50].

The described approaches to pseudo-targeted metabolomics are using two MS instruments, but Wang et al. managed to develop a new strategy for pseudo-targeted metabolomics that could be achieved on one LC/Q-TOFMS instrument (i.e., SYNAPT G2-Si Q-TOF MS; Waters Corp.) [51]. The MS is operated in the multiple ion monitoring (MIM) modes with time-staggered ion lists (tsMIM). Full scan-based untargeted analysis was applied to extract the target ions. After peak alignment and ion fusion, a stepwise ion-picking procedure was used to generate the ion lists for subsequent single MIM and tsMIM metabolite pseudo-quantitation. A hybrid Q-Orbitrap instrument was also used for metabolites profiling and quantitation (i.e., pseudo-targeted metabolomics on one platform) to reveal the effects of polyphenols extracted from different tea samples on the metabolic regulation of inflammatory response at the cellular level [52]. The acquired HRAM data in this study enabled the confident and accurate identification of targets of biological interest to measure the dynamic changes of metabolites. This led to the discovery that the treatment with selenium-enriched green tea, for example, can play an immune protective role at a lower concentration and be involved in three unique pathways of antioxidant enzyme activation, including phenylalanine, tyrosine and tryptophan biosynthesis, phenylalanine metabolism, and CoA biosynthesis [52].

#### 4.2.3. Semi-Targeted Metabolomics

Among other terms, “semi-targeted” has been heavily used by the community to describe the work combining the advantages of both targeted and untargeted metabolomics. However, this term is also used to describe different approaches that do not necessarily have the same aim. This might be misleading to users who are looking to develop a comprehensive metabolomics approach.

Dunn et al. have considered semi-targeted metabolomics as an approach for the confident chemical identification and structural elucidation of detected metabolites (i.e., qualitative) via the utilization of authentic chemical standards and methods that provide high accuracy, precision, and selectivity. Moreover, this approach is using MS data for semi-quantitation (relative quantitation) [22]. Interestingly, Li et al. have used the same approach of creating a plant metabolite spectral library using 544 authentic standards, which increased the efficiency of identification for untargeted metabolomic studies, but they called the approach a pseudo-targeted method [53]. A multi-platform (2D LC-MS/MS and NMR) protocol was also used for the aim of identification of natural compounds of the Mediterranean marine sponge *Crambe crambe*. In this protocol, the semi-targeted term was used to describe the level of identification, which benefited from using NMR, rather than increased metabolite recovery [54]. Similarly, other analyses were done for the confident annotation of lipids using HRAM [55], QTRAP [56], QTOF [57,58], or metabolites using DI-FT-ICR-MS coupled with pathway enrichment analysis [59], but authors still called these approaches semi-targeted lipidomics or metabolomics. In another example, MS^2^ spectra of metabolites resulting from a QTRAP instrument were compared with references MS^2^ spectra for the confident annotation of potential biomarkers of fertility decline in elderly women, which might help in finding the corresponding traditional Chinese medicine treatment and exploring its mechanism of action [60]. Moreover, a method that used tandem MS coupled to the capillary electrophoresis (CE) technique was used for improved annotation of unknown compounds utilizing both the MS/MS fragmentation patterns and CE migration time [61], yet the authors called the approach semi-targeted metabolomics. Bayle et al., on the other hand, performed a multi-platform approach using both liquid and gas chromatography coupled to MS to improve the coverage of the physiochemical diverse nutrient markers in plasma. They called their approach semi-targeted metabolomics since it enabled the quantitation of desired metabolites and the discovery of new features resulting from a metabolic change [62]. 

Improvements to the hybrid quadruple-orbitrap technologies regarding scan speed and sensitivity have been considered as a milestone that would enable an accurate and improved throughput semi-targeted analysis that combines both the quantitation and the discovery on a single platform without significant sacrifice of analytical fidelity [18,19]. Such improvements have enabled the profiling of compounds with largely varying abundances while maintaining high mass accuracy and resolving power. HRAM coupled to capillary ion chromatography was, for example, used for a powerful semi-targeted approach for the analysis of metabolites related to the energy metabolism pathways (e.g., di- and triphosphates and organic acids) that are poorly resolved with conventional metabolomics analytical techniques [63]. This method is called a semi-targeted analysis since it combined the quantitation of the selected metabolites using their authentic standards but also performs a discovery untargeted analysis on the acquired data. Similarly, an integrated semi-targeted metabolomics platform was incorporated for high throughput metabolite identification and quantitation to reveal distinct metabolic dysregulation in pleural effusion caused by tuberculosis and malignancy utilizing an orbitrap-based MS with DDA MS^2^ acquisition [25]. GC coupled to orbitrap MS also showed an ability to perform quantitative as well as qualitative analyses (called semi-targeted) to measure the differences in metabolite profiles in pigmented rice grains that might provide insight into the enhanced antioxidant capacities of those grains [64]. These studies demonstrated that the high selectivity of HRAM technology is crucial for the analysis not only of isobars but also of low abundant metabolites in complex backgrounds. 

Other MS platforms were also used to perform semi-targeted metabolomics in the sense of combining the outcome of both targeted and untargeted analysis from a single injection on TOF-based MS [27] or a two-injection strategy [65]. The latter example presents using triple quad MS data with UV detection and RT information (comparison with standards or data from literature) for metabolomics profiling followed by a second injection for relative quantification via SIM mode UPLC-MS analyses.

#### 4.2.4. Simultaneous Quantitation and Discovery (SQUAD) Analysis

Many of the approaches presented in previous sections consist of multiple injection strategies, either on different platforms or on the same platform. These multi-injections are practiced, normally, by performing untargeted profiling (preferably on an HRMS) to provide information on potential targets to be quantified via an MRM, SRM, or SIM mode in a second injection.

SQUAD analysis is a promising alternative, offering a way to strike the balance between untargeted and targeted approaches in one single experiment. The SQUAD workflow begins in much the same way as targeted approaches, where researchers annotate and quantify a pre-selected group of metabolites in a sample. However, the data can then be reanalyzed (or retro-mined) to look for global metabolic changes that were not part of the original focus (see Figure 3). SQUAD analysis can, therefore, identify other biologically meaningful metabolic changes that the scientist may not have been aware of in their signaling pathway of interest.

SQUAD analysis utilizes pooled mixes of the analyzed biological samples as QC samples to evaluate the stability and robustness of the analytical system, to be used for correction and normalization and to collect compound fragmentation information that could be used for the structure elucidation of unknown metabolites. Like untargeted metabolomics, it is crucial to acquire high-resolution and accurate mass data to enable confident unknown annotation in SQUAD analysis. In addition, HRMS mass spectrometers facilitate the resolution of low abundant metabolites in a complex matrix. Therefore, it is a good choice in this approach. The recent advancement in Orbitrap Tribrid mass spectrometers is a golden opportunity to perform a SQUAD analysis utilizing the sensitive linear ion trap for the quantitation of metabolites without sacrificing the discovery portion of the untargeted assay performed on the high-resolution accurate mass orbitrap analyzer (see Figure 4). This fast alternating eliminates the variability of using multiple instruments and the need to re-inject limited biological samples. Tribrid platforms enable various methods for MS^2^ fragmentation (e.g., HCD, CID, and UVPD) and even allow the MSn fragmentation capability that brings MS-based unknown identification to a higher accuracy level. Hybrid-based orbitrap instruments, on the other hand, have also improved their scanning speed, making polarity switching feasible even with high throughput LC and GC methods. This indeed allows higher recovery of the metabolome features once used for a SQUAD analysis, while still being able to have enough MS^1^ scans for peak quantification and perform MS^2^ fragmentation for structural elucidation. Samples can be profiled in both polarities in a single injection while the QC pools are used for deep fragmentation analysis individually. This increases throughput while maintaining both the targeted and nontargeted analysis integrity.

Other MS methodologies such as EAD (e.g., Sciex ZenoTOF) and ion mobility (e.g., Bruker timsTOF) can benefit the untargeted portion of the SQUAD analysis since they provide a tool for confident unknowns’ annotation. Similarly, the availability of multi-ionization techniques like ESI and APCI (LC-based), or EI and CI (GC-based) can enhance the breadth of analytes observed in the untargeted analysis.

UHPLC-MS-based SQUAD can be more popular than GC-MS-based workflows since LC-MS methods require less complicated sample preparation (e.g., no derivatization steps are required) and have an increased ability to identify and measure a broader range of compounds. However, this does limit the SQUAD approach to LC-MS only. GC-MS, for example, can be used to identify volatile and semi-volatile compounds for metabolic investigations. 

The quantitation part of the SQUAD workflow is flexible, and it is up to the user to define its accuracy level and the number of targets to be quantified. If both the isotopically labeled IS and unlabeled standards are used to make calibration curves, and the IS are spiked into the samples at known concentrations to correct for variations in ionization efficiency, then it is an absolute quantitation of metabolites. However, a dedicated IS for each metabolite of interest may not always be feasible due to high costs or limited availability. Standards only, thus, are used to make calibration curves to determine how signal intensity responds as a function of analyte concentration and the range of linearity of this relationship [66]. Finally, a relative quantitation (relative quantification with one-point calibration) with respect to a reference sample can also be achieved. It is crucial to recall that absolute quantitation, when feasible, is of a greater value since it enables measurement of the thermodynamics of metabolic reactions [67] and the molecular dynamics underlying the flow of atoms through a metabolic network [68]. The ability to compare results across instruments and laboratories is also more robust when doing absolute quantitation as ionization and instrumentation variabilities are adjusted using stable labeled coeluting standards.

It is important to mention that to increase the number of quantified metabolites in a SQUAD analysis, users should optimize the separation (e.g., LC or GC) and the mass detection methods accordingly. This is to ensure recording enough scans per peak (≥8 scans) for accurate peak integration, to secure sufficient sensitivity for trace amounts measurement, and to have a linear dynamic range covering the levels of targeted analytes. For example, lowering the orbitrap resolution from 240 k to 120 k on a hybrid system can significantly increase the number of scans per peak without sacrificing the quality of MS^1^ accuracy for the discovery portion. An optimized separation method, on the other hand, might enable isomers separation to be quantified via MS^1^ scanning rather than an MS^2^ method such as PRM.

### 4.3. Opportunities for SQUAD Analysis

One of the biggest strengths of SQUAD analysis is the ability to perform targeted and untargeted analysis in a single sample injection, which is particularly advantageous for laboratories that have limited access to samples, time, and resources. This also offers a powerful and efficient way to gain more knowledge from valuable biological samples. SQUAD analysis on HRMS platforms also allows discovery in clinical studies, since they record high-quality MS^1^ full scans and have improved the ability to perform sensitive and reliable quantifications of a large variety of analytes in MS^1^ full scans as well. The development of intelligence-driven data acquisition strategies enables scientists to dive deeper into the sample while providing an overview of known metabolites. Koelmel et al., for example, developed an LC-MS/MS DDA with an automated exclusion list (i.e., IE-Omics) for expanding lipidome coverage. A more advanced and intuitive method was developed by Thermo Fisher Scientific (i.e., AcquireX), which extends intelligence-driven mass spectrometry through experimental connectivity by integrating independent experiments into an automated workflow that enhances real-time, selective LC-MSn data acquisition for efficient and comprehensive sample and study characterization. With five different routines, the AcquireX data acquisition workflow extends productivity to all small-molecule applications, from comprehensive structural annotation to screening. These methods maximize the number of relevant compounds interrogated by MS^2^ and offer several benefits to users, including integrating independent experiments into automated workflows. This thereby increases the efficiency and ease of use of LC-MS-based SQUAD analysis. Agilent technologies developed a similar tool (i.e., iterative MS/MS) for improved metabolome coverage.

Comprehensive metabolites databases (e.g., mzCloud, NIST, Chemspider, KEGG, etc.) and improved multivariate data analysis methods and software (Compound Discoverer, SIMCA, etc.) have enabled the analysis of large quantities of metabolic profiling data (i.e., reveal clustering based on features). Open sources data analysis solutions such as XCMS [69], MZmine [70], MetAlign [71], MS-DIAL [72], and Skyline [73] are also available for automated processing of MS-based metabolomic data.

In addition to clinical applications, the SQUAD approach can also be used in bacterial metabolism analysis. The human gut microbiota plays an important role in human physiological processes such as nutrient digestion and the regulation of the immune system [5]. Therefore, measuring traditional microbiota metabolites (e.g., short-chain fatty acids and bile acids) in the gut and simultaneously performing a discovery-based analysis can provide a better understanding of the activity of gut microbes, and further inform us of their impact on human health. SQUAD can also be utilized toward microbial metabolism and pharmacomicrobiomics (i.e., study the interaction between xenobiotics, or foreign compounds, and the gut microbiome). This would provide information on how the individual gut microbiome informs drug efficacy in patient cohorts.

The metabolic engineering field can also benefit from SQUAD analysis. The new workflow could be used to identify metabolic pathway bottlenecks and to identify diagnostic biomarkers. Such biomarkers can be utilized to optimize microbial growth rate, increase strain tolerance, improve stress regulation and adaptation, improve substrate utilization and uptake, and increase product titer/rate/yield. At the same time, SQUAD will be providing quantitative information about the desired production like any traditional targeted approach.

SQUAD-based analyses are also expected to play an important role in the natural products (e.g., flavonoids, steroids, etc.) field, especially when performed on platforms that provide multiple fragmentation methods and solutions for intelligent MS fragmentations for unknown identification. Orbitrap Tribrid mass spectrometers offer both CID and HCD and the UVPD fragmentation method, which produces fragment ions indicative of double bond locations and other unique structurally diagnostic information for analysis of various compound classes, including lipids and glucuronides. Moreover, they offer real-time library search, which uses Real-Time MS^2^ spectral matching against spectral libraries for decision-based MSn triggering. This leads to higher confidence in metabolite annotation and improved characterization of unknowns. Unknowns’ annotation can also benefit from using EAD fragmentation, which can provide unique peaks for some lipid classes that allow for deeper characterization of individual lipid molecular species. Ion mobility also adds a new dimension to LC-MS untargeted metabolomics, which significantly enhances coverage, sensitivity, and resolving power for analyzing the metabolome, particularly metabolite isomers.

### 4.4. Barriers to Adopting SQUAD

Despite the clear advantages that SQUAD analysis offers for metabolomics; its adoption might be restricted by barriers that need to be addressed by vendors and researchers to enable its wider application in laboratories. The first barrier is the availability of pure and diverse chemical standards, labeled standards needed to perform confident identification, and, optionally, accurate absolute quantitation. These authentic standards are used to confirm the identity of metabolites by matching multiple properties such as separation retention time, accurate molecular mass, and MS fragmentation patterns. While many of these authentic metabolite standards are commercially available, many of them are not. The main barrier, thus, is the need for custom chemical synthesis, when true unknown compounds are identified, which significantly increases the cost. The untargeted part of SQUAD workflows, however, can help the community by providing insights toward important and relevant metabolites to be synthesized, which could then feed the targeted part of future SQUAD studies.

Second, high-quality data must be obtained to ensure a confident interpretation of the results. Therefore, choosing a platform with high mass accuracy, high resolution, and biological and statistical robustness is essential to ensure confident annotation of unknown metabolites and accurate differential analysis. Orbitrap-based MS and ToF-based MS with high-resolution mode can be utilized for this task but not the low-resolution mass spectrometers such as QQQ and quadrupole trap, with orbitraps being superior in this regard. Finally, data processing is another key component of SQUAD analysis, and retro-mining data requires powerful data analysis capabilities. These solutions should be able to enable fast data processing and analysis with accurate quantification of metabolites. They should also facilitate differential analysis and confident metabolite annotation utilizing high-quality spectral libraries and databases for accurate discovery analysis. In addition, such solutions should offer metabolic pathway analysis for accurate biological interpretation of acquired data.

Table 1 compares the traditional metabolomics workflows (i.e., untargeted and targeted) and the novel SQUAD analysis. This includes, for example, the differences between the aim, advantages, key applications and opportunities, and limitations of each workflow. Table 1 enables a quick guide that enables users to decide what workflow to adopt.

## 5. Conclusions

The recently developed SQUAD metabolomics technique is a promising alternative, offering researchers a new way to merge untargeted and targeted approaches in one single experiment. It also offers laboratories a new way to approach metabolomics. This workflow enables the annotation and quantification of a pre-selected group of metabolites in a sample. In addition, the data can then be reanalyzed (or retro-mined) to look for global metabolic changes that were not part of the original focus. This simultaneous acquisition of hypothesis-led and discovery-led datasets allows scientists to gain more knowledge about biological systems in a single experiment. Weighing the pros and cons of untargeted and targeted metabolomics has often held back the tremendous potential of metabolomics in life sciences research. Now, the SQUAD analysis provides the best of both worlds to unlock this potential for scientists worldwide.

## Figures and Tables

**Figure 1 metabolites-13-00648-f001:**
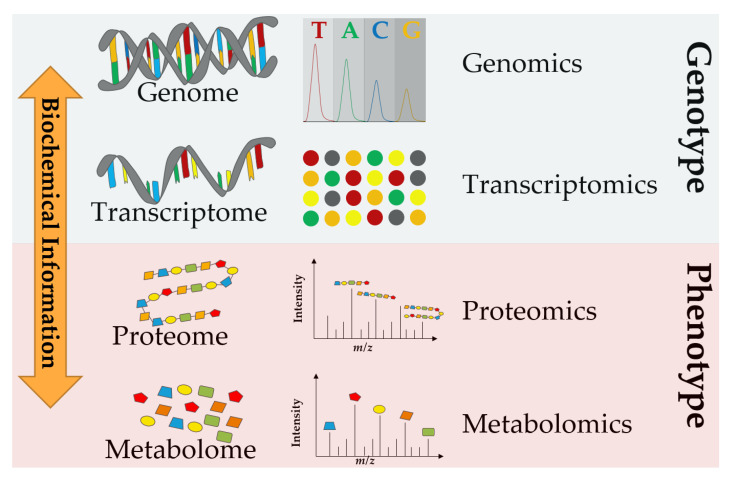
A multi-omics approach allows the measurement of the flow of molecular information from genes to metabolites to explain or predict phenotype from genotype.

**Figure 2 metabolites-13-00648-f002:**
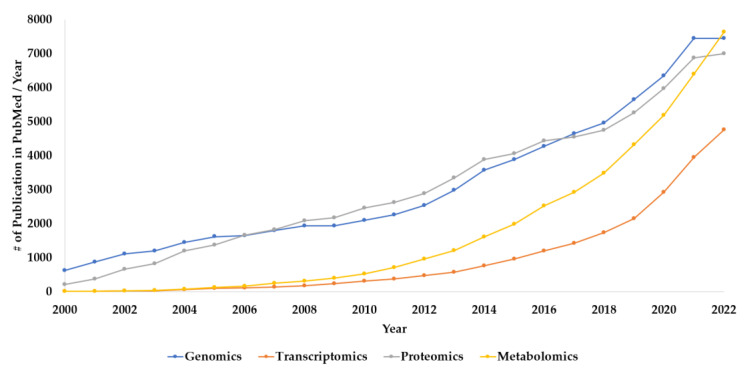
The number of annual publications that utilize “omics” technologies during the period of 2000–2022. Search Criteria: the individual “omics” technology was selected in “title” and “title/abstract”. The search was conducted using the PubMed platform.

**Figure 3 metabolites-13-00648-f003:**
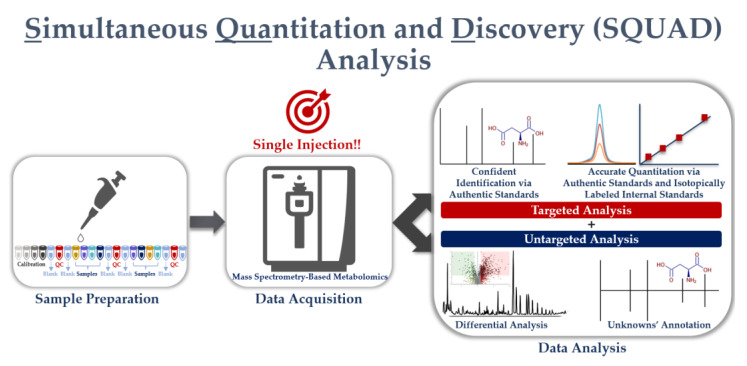
The novel single injection simultaneous quantitation and discovery (SQUAD) metabolomics combines targeted and untargeted workflows.

**Figure 4 metabolites-13-00648-f004:**
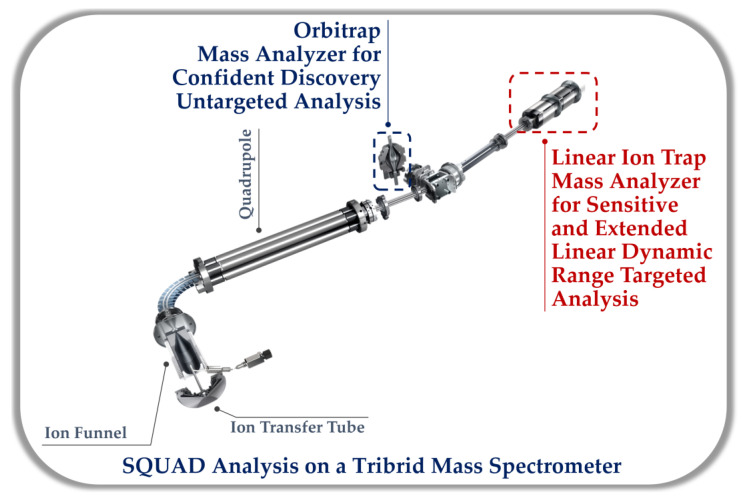
SQUAD analysis on an Orbitrap Tribrid mass spectrometer utilizing the sensitive linear ion trap for the quantitation of metabolites without sacrificing the discovery portion of the untargeted assay performed on the high-resolution accurate mass orbitrap analyzer.

**Table 1 metabolites-13-00648-t001:** A comparison of the traditional metabolomics workflows (i.e., untargeted and targeted) and the novel simultaneous quantitation and discovery (SQUAD) analysis.

Metabolomics Workflow	Untargeted	Targeted	SQUAD
Type	- Hypothesis-generating - Qualitative analysis	- Hypothesis-driven - Quantitative analysis	- Hypothesis-driven and generating - Both qualitative and quantitative
Study/aim	- Determine all measurable metabolites in a biological sample	- Measure defined groups of chemically characterized metabolites	- Both
Advantages	- Wider coverage of the metabolome - Leads to discovery	- Absolute and accurate quantitation of metabolites	- Combines quantitation and discovery
Key applications	- Research fields - Natural products discovery - Disease phenotyping, diagnosis, and mechanism	- Clinical routine diagnosis - Dietary assessment in nutritional metabolomics - Forensic toxicology	- Able to do all the mentioned applications in a single injection
Limitations	- No absolute quantitation data - Low annotation/identification accuracy in the absence of authentic chemical standards - Low coverage of diverse metabolites in a single method	- Information on a limited number of metabolites - Might require extra sample preparation steps	- Availability of pure chemical standards and internal standards - Not applicable on low-resolution triple quadrupole MS instruments
Key acquisition instrumentation	- Orbitrap-based MS - TOF MS	- Triple quadrupole MS	- Orbitrap-based MS - TOF MS
Opportunities	- Biomarkers discovery - Enable pathway analysis	- Biomarkers validation - Limited pathway analysis	- Biomarkers discovery and validation - Enable pathway analysis

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
