# Peer review of "Simultaneous Quantitation and Discovery (SQUAD) Analysis: Combining the Best of Targeted and Untargeted Mass Spectrometry-Based Metabolomics"

_metabolites, 2023, doi:10.3390/metabo13050648_

Round 1

Reviewer 1 Report

The manuscript submitted by Amer and colleagues is a review article which discusses different traditionally applied analytical strategies in metabolomics and a more modern analytical strategy (SQUID). The manuscript overall covers a need in the metabolomics community to provide a clear distinction between the traditional untargeted/targeted strategies and the hybrid approach which combines advantages of both of these strategies and I expect to be well read and cited. The manuscript is overly commercial in my opinion and neds to focus on the science and not the technologies because SQUID can be applied on any HRMS LC-MS (or GC-MS) system. The manuscript is also quite long for its topic/focus. However, I enjoyed reading the manuscript.

There are some items which the authors should consider in any revision of the manuscript.

1. Line 53/54. LC-MS instrumentation can be as expensive to purchase as a NMR system and so this statement should be modified or removed.

2. The paragraph starting at line 66 should be removed from the manuscript. This is very basic and any reader should already know this. There are other published works which demonstrate the expansion of omics techniques and these could be cited instead.

3. Line 91 onwards. The new SQUID technique should not be introduced before the traditional untargeted and targeted approaches are described because knowledge of the traditional approaches are required to understand what SQUID is. Therefore this paragraph and figure should be removed and moved to later in the manuscript.

4. The manuscript is very commercial with respect to one company but should be applicable to all different LC-MS and GC-MS systems. One example of this is found in lines 133-137 where only resources from one company (that of the authors are included). A more generic and wider description should be included to demonstrate both commercial but also open access resources in instances where there is a commercial theme (lines 133-137 is one of several instances of this in the manuscript).

5. The applications sections seem out of place in this review of analytical strategies and the application sections are not broad enough in relation to the large variety of scientific topics where metabolomics is applied. I would recommend that the applications sections are removed.

6. Line 199-200.This sentence reads as if the internal standards only are used in the dilution series which is not the case. This sentence should be modified to show that chemical standards (and not internal standards) are present in the dilution series.

7. Line 214-215. Why discuss semi-identification in a targeted assay section where semi-identification would not be a part of the process. This sentence should be removed.

8. Line 254. QA and QC is discussed in relation to targeted methods but is not discussed for untargeted and SQUID methods. Why? It is recommended that these topics should be discussed for all three analytical strategies.

9. Line 324. What do the authors mean by the term wide range calibration curves? This should be explained in more detail.

10. Line 380 and elsewhere. Where a person is named only used the surname and not the full name.

11. Line 533. What are pooled mixes as QC samples? Are these biological samples, chemical standard mixtures, SRM? Be more specific.

12. Figure 4 and related text defines that quantification is performed in the ion trap mass analyser of the tribrid instrument and full scan HRMS is applied in the Orbitrap mass analyser. Is SQUID only applicable on Thermo Fisher Scientific tribrid instruments or is it applicable on Thermo Fisher Scientific hybrid instrument (e.g. Exploris systems) and on Q_TOF systems. The more generic applications have to be discussed.

13. Line 565 onwards. How many metabolites are typically quantified in a SQUID method?

14. Line 572-573. This approach has to be explained in more detail, how can a single reference sample provide quantification of many metabolites? Is this relative or absolute quantification?

15. Line 672. I believe Professor Viant works at the University of Birmingham.

Author Response

The authors are glad to learn that the reviewer has found the SQUAD analysis as a more modern analytical strategy and that the manuscript overall covers a need in the metabolomics community to provide a clear distinction between the traditional untargeted/targeted strategies and the hybrid approach which combines advantages of both of these strategies. We are also extremely honored with the expectation that the manuscript to be well read and cited.

We do appreciate the feedback that the manuscript is “overly commercial and needs to focus on the science and not the technologies”, to this end, the manuscript was modified to ensure that it discusses the utilization of other MS technologies than orbitrap-based, other fragmentation methodologies such as EAD, and open sources tools for data analysis (e.g., XCMS, MZmine, MS-DIAL, etc.). Please see L582-585, L638-645, and L672-677 of the updated manuscript.

The authors agree with the reviewer that the manuscript is long, but our aim was to comprehensively cover the topic and make this review as a reference for the experts in the field but also for those whom new to the subject. Saying that, we considered the reviewer’s comment number 5 and deleted the two application sections for the untargeted and targeted analyses, which reduce the length of the manuscript.

We are also glad to learn that the reviewer has enjoyed reading the manuscript, which makes us proud that we managed to discuss the topic in a good quality review.

In addition, all revision items by the reviewer were addressed as described below:

1. Line 53/54. LC-MS instrumentation can be as expensive to purchase as a NMR system and so this statement should be modified or removed.

We agree with the reviewer that the cost of some LC-MS systems can be similar to an NMR instrument, therefore we removed the sentence as recommended. See L52-54 of the resubmitted manuscript.

2. The paragraph starting at line 66 should be removed from the manuscript. This is very basic and any reader should already know this. There are other published works which demonstrate the expansion of omics techniques and these could be cited instead.

The authors understand the reviewer’s point; however, this SQUAD review also aims to introduce metabolomics to new users, who have less background about the adoption of metabolomics as a family member of the “omics” family. On the other hand, the data in Figure 2 represent current data compared to what has already been published and is based on a recent unpublished tweet that has gained significant discussion in the community regarding the rapid increase of metabolomics publications in comparison to other “OMICS”. We feel it’s important to the integrity of the work to include the image.   

3. Line 91 onwards. The new SQUID technique should not be introduced before the traditional untargeted and targeted approaches are described because knowledge of the traditional approaches are required to understand what SQUID is. Therefore this paragraph and figure should be removed and moved to later in the manuscript.

This a valid point, we moved the section as recommended. Please see L332-338 of the resubmitted manuscript.

4. The manuscript is very commercial with respect to one company but should be applicable to all different LC-MS and GC-MS systems. One example of this is found in lines 133-137 where only resources from one company (that of the authors are included). A more generic and wider description should be included to demonstrate both commercial but also open access resources in instances where there is a commercial theme (lines 133-137 is one of several instances of this in the manuscript).

The authors agree with the reviewer on this point. Therefore, we added more examples of the commercially available but also the open sources MS systems, data analysis software, and metabolites databases. This includes EAD fragmentation technology by Sciex, ion mobility MS, iterative MS/MS software by Agilent Technologies, open sources software solutions such as XCMS, MZmine, MetAlign, MS-DIAL, and Skyline, and databases such as NIST, KEEG, and Checmspider. Please see L582-585, L638-645, and 672-677 of the resubmitted manuscript.

5. The applications sections seem out of place in this review of analytical strategies and the application sections are not broad enough in relation to the large variety of scientific topics where metabolomics is applied. I would recommend that the applications sections are removed.

We agree that the applications list is more than what is presented in this manuscript, but the aim was to present some of the common applications for both targeted and untargeted workflows. Since we agree with the reviewer that the manuscript is already long, and the application sections can be out of place in this review, we decided to remove them from untargeted and targeted sections as recommended by the reviewer. Since this review is meant to provide a “where to start” guide for new metabolomics users as well, we added a table at the end of the review that compares the traditional metabolomics workflows (i.e., untargeted and targeted) and the novel simultaneous quantitation and discovery (SQUAD) analysis. This table includes, for example, the differences between the aim, advantages, key applications and opportunities, and limitations of each workflow. See L705-710 of the resubmitted manuscript.

 “6. Line 199-200. This sentence reads as if the internal standards only are used in the dilution series which is not the case. This sentence should be modified to show that chemical standards (and not internal standards) are present in the dilution series.

The reviewer is absolutely correct. The sentence was modified accordingly. See L209-213 of the resubmitted manuscript.

7. Line 214-215. Why discuss semi-identification in a targeted assay section where semi-identification would not be a part of the process. This sentence should be removed.

The sentence was removed as recommended.

8. Line 254. QA and QC is discussed in relation to targeted methods but is not discussed for untargeted and SQUID methods. Why? It is recommended that these topics should be discussed for all three analytical strategies.”

The authors appreciate this valuable comment. The adoption of QA/QC strategies is crucial for the improvement of the integrity, quality, and robustness of targeted, untargeted, and SQUAD metabolomics. A new paragraph was added to the untargeted metabolomics section, see L128-139. The utilization of pooled QC samples for the evaluation of the stability and robustness of the analytical system, for correction and normalization, but also to collect compound fragmentation information that could be used for the structure elucidation of unknown metabolites in SQUAD analysis was already highlighted in the submitted manuscript, see L555-557 of the resubmitted manuscript.

9. Line 324. What do the authors mean by the term wide range calibration curves? This should be explained in more detail.

The reviewer is right, this sentence is unclear. This is due to a typo, it should be as: “to relative quantitation in the absence of IS and calibration curves (i.e., one-point calibration),”. Please see corrected sentence L344-345 of the resubmitted manuscript.

10. Line 380 and elsewhere. Where a person is named only used the surname and not the full name.

This was corrected accordingly, see L402, 409, and 426 of the resubmitted manuscript

11. Line 533. What are pooled mixes as QC samples? Are these biological samples, chemical standard mixtures, SRM? Be more specific.”

The QC samples in this case are mixtures of the analyzed biological samples since they represent all analytes detected in those samples. Please see modified sentence L555-556 of the resubmitted manuscript.

“12. Figure 4 and related text defines that quantification is performed in the ion trap mass analyser of the tribrid instrument and full scan HRMS is applied in the Orbitrap mass analyser. Is SQUID only applicable on Thermo Fisher Scientific tribrid instruments or is it applicable on Thermo Fisher Scientific hybrid instrument (e.g. Exploris systems) and on Q_TOF systems. The more generic applications have to be discussed.

The authors thank the reviewer for this important and critical question. SQUAD analysis is applicable on any full scan MS instrument. This includes hybrid Thermo instrumentations (please see L570-577), but also other vendors MS systems such as ToF like Agilent Q-TOF, Bruker timsTOF, and Sciex ZenoTOF. The advantage of Thermo Tribrid systems is that they provide a parallel analysis where a sensitive quantitation with an extended linear dynamic range can be performed using the linear ion trap, and a high-resolution accurate mass analysis for the untargeted discovery is performed using the orbitrap. The manuscript was modified to clarify this point. See L582-585.     

13. Line 565 onwards. How many metabolites are typically quantified in a SQUID method?”

This is an important question. The number of quantified metabolites depends on the application itself, but in general, any SQUAD assay depends on 1) number of scans per peak for accurate peak integration, 2) high sensitivity for trace amounts measurement, 3) enough linear dynamic range, and 4) the availability of authentic standards and IS for confident absolute quantitation. In any SQUAD analysis, the separation technique (e.g., LC or GC) and the MS method should be optimized to secure points 1-3, especially when increasing the number of quantified targets. For example, lowering the orbitrap resolution from 240k to 120k on a hybrid system can significantly increase the number of scans per peak without scarifying the quality of MS1 accuracy for the discovery portion. An optimized separation method, on the other hand, might enable isomers separation to be quantified via MS1 scanning rather than an MS2 method such as PRM. Finally, performing a parallel analysis using a Tribrid can elevate the number of quantified targets on the ion trap (sensitive and extended linear dynamic range) without sacrificing the discovery analysis on the orbitrap (accurate MS). Overall, this is why the authors stated in the manuscript that “The quantitation part of the SQUAD workflow is flexible”. To this point, we modified the manuscript to clarify this point. Please see L607-616.     

14. Line 572-573. This approach has to be explained in more detail, how can a single reference sample provide quantification of many metabolites? Is this relative or absolute quantification?

Similar to question 9. This was clarified in the manuscript. The authors mean one-point calibration. See L600.

15. Line 672. I believe Professor Viant works at the University of Birmingham.”

The authors thank the reviewer for this correction. It was corrected accordingly.

Reviewer 2 Report

The authors discuss mass spectrometry-based metabolomics approaches with focus on simultaneous quantitation and discovery (SQUAD) analysis. Given the increasing application of metabolic analyses in a variety of research fields, this review is interesting and can be considered for publication after some improvements.

1. It would be help the readers if the review includes a table summarizing applications, advantages, limitations, and apparatus for untargeted, targeted and SQUAD approaches.

2. Section 4.2 and subsection 4.2.4 have the same title. And other approaches than SQUAD descried in section 4.2 do not use single injection on the same platform, so they should not be SQUAD. To avoid confusion, the authors may either modify the title for section 4.2 or discuss those approaches using multiple injection strategies in a separate section.

3 Figure 3 may be placed in the SQUAD section, but this is optional.

Author Response

The authors are grateful that the reviewer finds this manuscript an interesting review and can be considered for publication. All of the reviewer’s comments were addressed as shown below.

1. It would be help the readers if the review includes a table summarizing applications, advantages, limitations, and apparatus for untargeted, targeted and SQUAD approaches.”

The authors appreciate this suggestion. Table 1, which compares the traditional metabolomics workflows (i.e., untargeted and targeted) and the novel simultaneous quantitation and discovery (SQUAD) analysis was added to the review. This includes, for example, the differences between the aim, advantages, key applications and opportunities, and limitations of each workflow. Table 1 enables a quick guide that enables users to decide what workflow to adopt. See L705-709 of the resubmitted manuscript.

2. Section 4.2 and subsection 4.2.4 have the same title. And other approaches than SQUAD descried in section 4.2 do not use single injection on the same platform, so they should not be SQUAD. To avoid confusion, the authors may either modify the title for section 4.2 or discuss those approaches using multiple injection strategies in a separate section.

We believe that the reviewer means the title of sections 4 and 4.2.4, which is “Simultaneous Quantitation and Discovery (SQUAD) Analysis”. We thank the reviewer for this comment; however, we feel both section titles should remain. The intention of this review is to educate and push the community to use the name SQUAD instead of other names in order to align the understanding around the overall complexity and integrity of the experiment. Section 4 naming must state SQUAD to introduce and solidify this concept.

3 Figure 3 may be placed in the SQUAD section, but this is optional.”

Figure 3 was moved to the SQUAD section as recommended. Please see L332-338.

Reviewer 3 Report

The manuscript metabolites-2384970 entitled Simultaneous Quantitation and Discovery (SQUAD) Analysis: Combining the Best of Targeted and Untargeted Mass Spectrometry-Based Metabolomics

by Bashar Amer and coworkers present a novel single injection simultaneous quantitation and discovery (SQUAD) metabolomics that combines targeted and untargeted workflows. It is used to identify and accurately quantify a targeted set of metabolites. It also allows data retro-mining to look for global metabolic changes that were not part of the original focus. This simultaneous acquisition of hypothesis-led and discovery-led datasets allows scientists to gain more knowledge about biological systems in a single experiment.

This review work extensively reviewed the work in the field of metabolomics.

The manuscript is well written, figure of quality and informative.

Minor point

Line 459: the title is at the endo of the page and should be at the beginning of the section

A minor linguistic revision is racomanded

Minor revision required

Author Response

“This review work extensively reviewed the work in the field of metabolomics. The manuscript is well written, figure of quality and informative.”

The authors are grateful to learn that the reviewer finds this manuscript as an extensive review and of good quality. All of the reviewer’s points were addressed as shown below.

“Line 459: the title is at the endo of the page and should be at the beginning of the section”

The authors are aware of this point, this issue will be fixed during the final editing steps once the manuscript is accepted for publication.

A minor linguistic revision is recommended

We appreciate the reviewer’s feedback. The authors, including author SB who is a native English speaker, have revised the language of the manuscript and corrected it accordingly.

Reviewer 4 Report

The review article summarized "Simultaneous Quantitation and Discovery (SQUAD) Analysis: Combining the Best of Targeted and Untargeted Mass Spectrometry-Based Metabolomics". In my opinion, the content of the review was presented very clearly to the readers. 

Author Response

The review article summarized "Simultaneous Quantitation and Discovery (SQUAD) Analysis: Combining the Best of Targeted and Untargeted Mass Spectrometry-Based Metabolomics". In my opinion, the content of the review was presented very clearly to the readers.

The authors are grateful to learn that the reviewer thinks this is a clearly presented review.